Photosynthetic responses of Halimeda scabra (Chlorophyta, Bryopsidales) to interactive effects of temperature, pH, and nutrients and its carbon pathways

Zuñiga-Rios Daily
Vásquez-Elizondo Román Manuel
Caamal Edgar
Robledo Daniel daniel.robledo@cinvestav.mx
Department of Marine Resources, Cinvestav , Merida , Yucatan , Mexico
Pawlik Joseph
Electronic publication date: 2021 Mar 5
Publication date: 2021
Volume: 9
Electronic Location ID: e10958
Received 2020 Nov 12; Accepted 2021 Jan 27
Copyright: ©2021 Zuñiga-Rios et al.
Copyright year: 2021
Copyright holder: Zuñiga-Rios et al.
License: This is an open access article distributed under the terms of the Creative Commons Attribution License, which permits unrestricted use, distribution, reproduction and adaptation in any medium and for any purpose provided that it is properly attributed. For attribution, the original author(s), title, publication source (PeerJ) and either DOI or URL of the article must be cited.
License URL: https://creativecommons.org/licenses/by/4.0/

Keywords: Carbonic anhydrase, CCM, 13C isotope, Halimeda scabra, Interactive effects, Nutrients, pH, Photosynthesis, Q10, δ13C

Funding: CONACYT Ph.D. scholarship 704918/593231 Project FOMIX-YUC-2014-C17-247043 CONACYT Program “Estancias Posdoctorales Nacionales 2019, 2020” 206050 This work was supported by CONACYT Ph.D. scholarship No 704918/593231, project FOMIX-YUC-2014-C17-247043 and a postdoctoral fellowship (No 206050) from the CONACYT Program “Estancias Posdoctorales Nacionales 2019, 2020”. The funders had no role in study design, data collection and analysis, decision to publish, or preparation of the manuscript.

==============================
In this study, we evaluated the interactive effects of temperature, pH, and nutrients on photosynthetic performance in the calcareous tropical macroalga Halimeda scabra. A significant interaction among these factors on gross photosynthesis (Pgross) was found. The highest values of Pgross were reached at the highest temperature, pH, and nutrient enrichment tested and similarly in the control treatment (no added nutrients) at 33 °C at the lowest pH. The Q10 Pgross values confirmed the effect of temperature only under nutrient enrichment scenarios. Besides the above, bicarbonate (HCO3−) absorption was assessed by the content of carbon stable isotope (δ13C) in algae tissue and by its incorporation into photosynthetic products, as well as by carbonic anhydrase (CA) inhibitors (Acetazolamide, AZ and Ethoxyzolamide, EZ) assays. The labeling of δ13C revealed this species uses both, CO2 and HCO3− forms of Ci relying on a CO2 Concentration Mechanism (CCM). These results were validated by the EZ-AZ inhibition assays in which photosynthesis inhibition was observed, indicating the action of internal CA, whereas AZ inhibitor did not affect maximum photosynthesis (Pmax). The incorporation of 13C isotope into aspartate in light and dark treatments also confirmed photosynthetic and non-photosynthetic the HCO3−uptake.

Introduction

Photosynthetic parameters respond faster to environmental changes than algae C and N content, hence their usefulness in short-term studies (Figueroa et al., 2009). Photochemical and biochemical reactions of photosynthesis continually respond to environmental conditions. Irradiance, temperature, and nutrient concentration including CO2 levels are among the main environmental factors limiting photosynthesis (Raven & Hurd, 2012; Zweng, Koch & Bowes, 2018). Algal ecophysiology studies have traditionally quantified temperature dependence using the metabolic quotient Q 10, which describes the metabolic increase accompanied by an increase of 10 °C in an optimal temperature range (Bruno, Carr & O’Connor, 2015; Vásquez-Elizondo & Enríquez, 2016). This quotient Q10 has also been used as a proxy to analyze the effect of temperature on nutrient absorption where it was found that by doubling the temperature the rate of nutrient absorption is doubled (Harrison & Hurd, 2001).

For aquatic plants, another limiting factor for photosynthesis is CO2, since it is the only source of carbon that can be assimilated by the Ribulose 1,5 bisphosphate carboxylase oxygenase enzyme (RuBisCO) (Falkowski & Raven, 2007). At seawater pH (8.1–8.3) CO2 is only between 0.5–1% of all dissolved inorganic carbon, while more than 91% is in the form of HCO3− and the remaining 8% is in the form of CO32− (Hurd et al., 2009; Diaz-Pulido et al., 2016). Moreover, since the diffusion of CO2through the cell membrane is slower in water than in air; many algae and higher plants have acquired mechanisms that promote intracellular CO2 accumulation, allowing photosynthetic organisms to reduce carbon limitation by increasing the concentration of CO2 in the vicinity of RuBisCO (CO2 Concentration Mechanisms, CCM). Parallel to this, CCM’s contribute to decreasing photorespiration due to the oxygenase activity of RuBisCO (Ogren, 1984; Enríquez & Rodríguez-Román, 2006; Cornwall, Revill & Hurd, 2015). In general, most algae can acquire inorganic carbon (Ci) for RuBisCO through diffusion and active absorption of both, CO2 and HCO3− (Badger & Price, 1994; Giordano, Beardall & Raven, 2005; Hurd et al., 2009). In many cases, the activity of CCMs has been associated with the direct or indirect use of HCO3− (Reiskind, Seamon & Bowes, 1988; Invers et al., 2001; Enríquez & Rodríguez-Román, 2006). Some macroalgae convert bicarbonate (HCO3−) into CO2 extracellularly using carbonic anhydrase (CA) thus CO2 enters the cell by active transport or diffusion. Other algae incorporate HCO3− actively through the cell membrane and, intracellularly, an internal CA converts HCO3− into CO2 (Badger & Price, 1994). The activity of carbonic anhydrases has been widely documented in algae (Reiskind, Seamon & Bowes, 1988; Invers, Perez & Romero, 1999; Enríquez & Rodríguez-Román, 2006) and they play a significant role in CCMs.

Many studies have examined the combined effects of environmental variables on algae photosynthetic responses: CO2 and temperature (Campbell et al., 2016; Kram et al., 2016; Vásquez-Elizondo & Enríquez, 2016); CO2 and light (Vogel et al., 2015); light and nutrients (Zubia, Freile-Pelegrín & Robledo, 2014); CO2 and nutrients (Hofmann et al., 2014; Hofmann et al., 2015; Bender-Champ, Diaz-Pulido & Dove, 2017); CO2, nutrients, and temperature (Stengel et al., 2014), and CO2, nutrients and light (Celis-Plá et al., 2015). Multiple stressors could have an interactive influence causing complex responses at the physiological and ecological level (Hofmann et al., 2014), which makes them difficult to interpret. Therefore, studies that combine ocean acidification scenarios with other factors such as temperature, light, and nutrient availability are particularly necessary since changes in these parameters are co-occurring with changes in carbonate chemistry in the seawater (Harley et al. , 2012; Hofmann et al., 2014).

Halimeda is a calcifying genus of siphonous green algae (Bryopsidales, Chlorophyta) which are important components of tropical and subtropical reefs and lagoons. Some species of this genus often appear dominating Caribbean coral reefs (Beach et al., 2003; Hofmann et al., 2014) where they contribute as primary producers, food source and habitat, sand production, and coral-reef formation. Halimeda scabra Howe is particularly abundant in the front reef and shallow rocky areas of the Caribbean Reefs (Alcolado et al., 2003). Despite the ecological studies above-mentioned, to our knowledge, no previous physiological studies have been reported for this species.

Photosynthetic responses to the combined effect of environmental variables have been studied in some Halimeda species, for example in H. opuntia, the effect of nutrients and pH (Hofmann et al., 2014; Hofmann et al., 2015), in H. incrassata and H. simulans the effect of pH and temperature (Campbell et al., 2016), and in H. opuntia the effect of pH and light (Vogel et al., 2015). These studies have suggested that an increase in both, CO2 (low pH) and high temperature could have a positive synergistic effect on photosynthetic rates (Kram et al., 2016). However, Halimeda responses to high CO2 have been diverse; in some species a decrease in photosynthesis with the reduction of pH has been observed (Price et al., 2011; Sinutok et al., 2012; Meyer et al., 2016) others have shown the opposite effect (Peach, Koch & Blackwelder, 2016) or a lack of a significant response (Price et al., 2011; Campbell et al., 2016). In general, there are still insufficient studies on the physiology of the genus Halimeda that allow us to understand the diversity of physiological responses to the interactive effects of environmental variables and the mechanisms involved in those responses.

In this study, we hypothesized that a synergistic increase in environmental factors (temperature, pH, and nutrients) would enhance H. scabra photosynthesis, which absorbs bicarbonate supported by a CCM. We evaluated the interactive effect of temperature, pH, and nutrient levels on photosynthetic responses of H. scabra. Additionally, we determined the Ci uptake mechanisms by measuring the effect of CA inhibitors on Pmax, analyzing δ13C values, and evaluating the incorporation of stable isotope 13C into resulting products of photosynthesis.

Materials and Methods

Biological material and culture conditions

H. scabra was collected in February 2017 in Xcalacoco, Quintana Roo, Mexico (20.660035 N, −87.034655 W), where it grows over rocky substrates between 1.5 and 2.0 m depth. In this area there are two marked seasons: a dry season from November to May with mean seawater temperatures of ∼24 °C and a rainy season from June to October, with mean ∼30 °C reaching extreme values of 33 °C, coinciding with summer, while the mean annual seawater temperature is ∼28 °C (Robledo & Freile-Pelegrín, 2005; Álvarez Cadena et al., 2007; Rodríguez-Martínez et al., 2010). The area is also characterized by submarine groundwater discharges to the coastal environment a pathway for nutrients transport from land to the marine environment (NO3−, NO2−, NH4 +, SRP, SRSi) conferring a pulsatile performance depending on the season, which can be particularly high in some localities especially during rainy season with pH important variations ranging from 7.0 to 8.5 (Crook et al., 2012; Hernández-Terrones et al., 2015).

Taxonomic determination of specimens was done according to Hillis-Colinvaux (1980). The algae were transported to the laboratory in cool boxes. At the laboratory, samples were cleaned with seawater to remove epiphytes and placed in 12 L aquarium with filtered seawater (36 PSU, pH 8.2) and kept under constant aeration at 24 °C. Irradiance was set at 115 µmol photons m−2 s−1 provided by fluorescent lamps under a 12:12 h light-dark photoperiod.

Photosynthetic measurements

To test the effect of temperature, pH, and nutrient levels on H. scabra photosynthesis, a three-factorial design with 36 combinations was used (Zar, 1996) (Table S1). The following treatments and levels were tested: (1) temperature at three levels (24, 28 and 33° C) maintained constant by placing the BOD bottles in a water bath connected to a temperature controlled water recirculation system (Cole-Parmer® Polystat® Refrigerated Recirculator, USA); (2) pH at three levels (7.5, 8.2, and 8.6) obtained by the addition of 0.5 M HCl or 0.5 M NaOH solutions (Lignell & Pedersen , 1989; Invers et al., 2001; Zou, 2014); (3) nutrient concentrations (KNO3:K3PO4) evaluated at four levels: low (1:0.1 µM), medium (5:0.5 µM), high (10:1.0 µM) and (4) control treatment without nutrients added to seawater. These temperature, pH and nutrient levels were selected according to prevailing conditions at the collecting site, as described above. To prepare the different combinations first seawater pH was adjusted, and 48 h later the pH was measured again and readjusted when necessary, after that the nutrients were added according to the required nutrient concentration (Lignell & Pedersen , 1989).

Photosynthetic responses were evaluated by the light-dark bottles method following the oxygen evolution versus irradiance (P-E curves) according to Thomas (1988) using a YSI 5000 Dissolved Oxygen Meter with YSI 5905 BOD Probe (YSI Incorporated Yellow Springs, OH, USA). To minimize wound effects, thalli were cut off and weighed 24 h before oxygen determinations. Apical fragments (0.1 g wet weight) were placed in 60 ml Biological Oxygen Demand (BOD) bottles containing the seawater previously prepared according to each of the 36 combinations. For each treatment seven bottles (n = 7) were used plus one blank, bottle filled with seawater only. Each combination was assessed on separated days with different fragments of algae tissues and according to each temperature level established in Table S1.

Once the temperature was set to the corresponding treatment the algae were exposed during one hour to each of seven successive irradiances selected (0, 100, 170, 200, 272, 436, 770 µmol photons m−2 s−1) generated by a 500 W halogen lamp and using different mesh size filters until darkness. Irradiances were measured with a spherical underwater quantum sensor (LI-193SA) connected to LI-1500 Light Sensor Logger (LI-COR, Nebraska, USA). The maximum photosynthesis rate (Pmax) was calculated as the average of the three highest oxygen production values at saturation irradiances. The dark respiration rate (Rd) was determined as oxygen consumption in total darkness, while the gross photosynthesis (Pgross) was determined as net photosynthesis plus dark respiration). At the end of each assessment the dry weight (DW) was determined, whereby the results were expressed as mg of oxygen g dry weight h−1 in 300 ml. All determinations were performed using Instant Ocean® synthetic seawater (Marineland, Blacksburg, VA, USA), prepared using distilled water and sterilized by autoclaving, this water was free of nitrate and phosphate therefore nutrient were added accordingly to the levels of each experimental treatments (Table S1).

Effect of temperature on Pgross: photosynthetic Q10 coefficient

To better understand photosynthetic responses to temperature we calculated the photosynthetic quotient Q10 of Pgross under different pH and nutrient conditions. The photosynthetic quotient was determined as the change in the photosynthetic rate within a rise in temperature of 5 °C, from 28 °C (T1) to 33 °C (T2) according to the following formula:

Q 10 = (Rate 2/Rate 1)(10∕T2−T1)

where, Rate 1 and Rate 2 were reaction rates measured at temperatures T1 and T2, respectively (Wernberg et al., 2016).

Inorganic carbon pathways

Bicarbonate (HCO3−) uptake for photosynthesis were assessed through three techniques: (1) carbon stable isotope (δ13C) values in algal tissue (2) CA inhibitor effects on Pmax, and (3) 13C stable isotope uptake and its incorporation into resulting products of photosynthesis.

Carbon stable isotope (δ13C) values in tissue from field samples

Whole thalli were carefully washed and decalcified in hydrochloric acid (HCl) at 0.6 M for 8 h, with hourly changes until full bubbling cessation. Afterward, the material was rinsed with distilled water and dried for 24 h at 70 °C. The dried material was ground in a mortar and sieved. Samples of five mg were weighed on analytical balance (precision of 0.0001 g) and individually packaged in microcapsules (5 ×  9 mm) for mass spectrophotometer isotopic analysis in the Stable Isotropy laboratory at the University of California at Davis, CA, USA.

Carbonic anhydrase inhibition assays

Two CA inhibitors were used in this study: (a) dextran-bound acetazolamide (AZ) that does not penetrate into the cell and only inhibits extracellular CA (Bjork et al., 1992), and (b) 6-ethoxyzolamide (EZ) that penetrates through the cell wall and membranes, and inhibits both external and internal CA (Bjork et al., 1992). AZ and EZ were dissolved in 0.05 N NaOH to a final concentration of 0.1 g ml−1 and 10 mM respectively (Bjork et al., 1992). Experimental treatments were prepared with seawater from the collecting area filtered with vacuum pump and sterilized by autoclaving. The inhibitors were added to the experimental seawater before the incubations to obtain a final inhibitor concentration of 100 µM (Bjork et al., 1992). Photosynthesis rates were tested under four treatments: (1) addition of AZ; (2) the addition of EZ; (3) the combination of both, AZ and EZ and (4) a control treatment with seawater without inhibitors. Maximum photosynthesis (P-E curves) was measured as previously described but at 28 °C of temperature (n = 7).

13C Labeling for the incorporation of NaH13CO3 into photosynthetic products

Initially, inorganic carbon was removed from filtered and sterilized seawater by reducing pH to ∼4 adding HCl 0.5 M and nitrogen bubbling for 5 h, subsequently the pH was raised to 8.2 adding NaOH 0.5 M (Invers et al., 2001; Zou, 2014). Afterward, 1.6 g L−1 of NaH13CO3 (isotope 13C 99% Aldrich) was added. H. scabra thalli fragments (2 g wet weight) were placed in hermetically sealed 250 mL BOD bottles (n = 3) containing seawater previously prepared with 13C isotope and maintained for 24 h at 28 °C of temperature under light saturation (278 µmol photons m−2 s−1, previously determined as H. scabra saturation irradiance, Ik (ratio of Pmax/ α, where α is photosynthetic efficiency). Three photoperiod treatments were selected: (1) 24 h in light, (2) 12:12 h light:darkness, and (3) 24 h in darkness. A control bottle containing seawater without 13C isotope was used in each treatment. At the end of the incubations the algae were washed with seawater and rinsed with distilled water to remove the remains of the isotope that were not absorbed, and later frozen and lyophilized. Lyophilized samples (0.6 g) were depigmented twice in succession with methanol (100%) after that, low molecular weight carbohydrates were extracted in distilled water for 24 h. Finally, the supernatant was frozen and lyophilized to be used in the NMR analysis.

13C-Nuclear Magnetic Resonance Spectroscopy (NMR) analyses

To determine the incorporation of NaH13CO3 isotope in photosynthetic products, lyophilized samples (8 mg) were dissolved in one mL with 99.8% deuterium oxide (D2O). The proton (13C) spectra were recorded on a Varian/Agilent Premium Compact 600 NMR spectrometer (Palo Alto, CA, USA) at a frequency of 150.83 MHz using Sodium [3-trimethylsilyl 2,2′,3,3′-2-H4] propionate (TSP-d4) with internal reference to 0.00 ppm. All NMR spectra were recorded at room temperature using the following parameters: scans = 50,000; 13C-pulse width of 3.3 s, an acquisition time of 0.5 s, and a relaxation delay of 0.60 s.

Statistical analyses

To test the interactive effects of temperature, pH, and nutrient levels on dependent variable (Pgross), a three-way ANOVA (3 × 3 × 4) was performed considering the three environmental factors as independent random variables. A Two-way ANOVA analyzed the effect of pH and nutrients on Q 10 Pgross. One-way ANOVA was applied to test differences between different inhibitor assays. Newman-Keuls post-hoc multiple comparisons were used to test significant differences among treatments. All statistical tests and analyses were performed using the statistical package Statistica™ 7. Before analyses, homogeneity of variance (Bartlett) and normality test (Kolmogorov–Smirnov) were tested, and transformations were applied if necessary.

Results

Photosynthetic responses to the interactive effect of temperature, pH, and nutrient levels

The three-way ANOVA showed a significant interactive effect of temperature, pH, and nutrients on H. scabra gross photosynthesis, Pgross (F12;216 = 4.57, p ≤ 0.001) (Fig. 1; Table S2). The highest Pgross values (1.83 mg O2 g DW h−1) were obtained at the highest nutrient concentration (10:1.0 µM) under elevated temperature (33 °C) at a pH of 8.6 and in the control treatment (no added nutrients) at 33 °C but, at the lowest pH (7.5) (Pgross = 1.78 mg O2 g DW h−1). In contrast, low and medium nutrient level treatments had lowest Pgross at intermediate temperatures. In general, H. scabra photosynthetic rates were higher at the highest temperature tested regardless of the nutrient or pH levels, except for the low nutrient treatment, which had higher Pgross at the lowest temperature for all pH treatments.

Figure 1 Effect of temperature, pH, and nutrients on the gross photosynthesis rate (Pgross) of H. scabra.

Interactive effect of temperature, pH, and nutrients on the gross photosynthesis rate (Pgross) of H. scabra (n = 7). Symbols represent the mean and error bars 0.95 confidence intervals.

It is noteworthy how Pgross responds to temperature and pH changes in relation to nutrient concentrations in an opposite pattern between the highest nutrient concentration (10:1.0) and the control treatment (no added nutrients) at the highest temperature tested.

Individual analysis of factors showed that pH alone had no effect on Pgross (F = 2.84, p >0.05), whereas the individual effect of nutrients (F = 4.77, p ≤ 0.05) and temperature (F = 45.30, p ≤ 0.001) were significant (Table S2). All interactions involving two factors were significant: temperature-pH (F = 2.70, p ≤ 0.05); nutrient-temperature ( F = 10.32, p ≤ 0.001); and pH-nutrients (F = 9.23, p ≤ 0.001).

Effect of temperature on Pgross (Q10Pgross) under nutrient concentrations and pH levels

The two-way ANOVA showed a significant effect of nutrient levels on Q10 Pgross (F2,54=6.721, p = 0.002). This effect was more pronounced with the highest nutrient concentration (Q 10 = 2.42) and decreased gradually as nutrient concentration decreased, from 1.75 to 0.75 in medium and low nutrient concentration, respectively. Conversely, pH and its interaction with nutrient levels did not show any significant effect on the Q10 calculated values (Table 1).

Table 1 Effect of temperature on Pgross (Q10) in H. scabra.

Effect of temperature on Pgross (Q10) in three nutrient concentrations and three pH levels (two way-ANOVA).

	Mean Q10	SS	DF	MS	F	p	
Nutrient ratio		10.839	2	5.419	6.721	0.002 ∗∗∗	
1.0:0.5	0.75b						
5.0:0.5	1.75a						
10.0:1.0	2.42a						
pH		3.137	2	1.568	1.945	0.152 ns	
nutrients ∗ pH		6.111	4	1.528	1.895	0.125 ns	
Error		43.543	54	0.806			
Notes.

ns not significant

*** significant

Letters indicate significant differences among mean based on Neuman-Keuls post hoc test.

HCO3− Uptake

The δ13C value in H. scabra was −23.9‰ suggesting uptake of both HCO3− and CO2, and the presence of a CCM. The carbonic anhydrase assays corroborate the latter since the addition of EZ caused a significant inhibition (22.2%) of maximum photosynthesis rates Pmax (F3,24 = 18.674, p ≤ 0.001) whereas the combination of both inhibitors produced a similar effect to that found with EZ (Fig. 2). AZ inhibitor showed no effect on Pmax implying a lack of external CA and direct uptake of HCO3− with a CCM depending on internal CA.

Figure 2 Carbonic anhydrase inhibitors on H. scabraPmax.

Comparison of the effect of two carbonic anhydrase inhibitors on Pmax percentage. Error bars represent confidence intervals at 0.95 (n = 7).

Incorporation of 13C into products of photosynthesis

13C isotope labeling in H. scabra observed by signal multiplicity (coupling) also showed bicarbonate uptake, since 13C isotope was incorporated into an amino acid akin to aspartate in the three photoperiod treatments analyzed. The incorporation in darkness indicates non-photosynthetic β-carboxylation. Aspartate also appears in the three control treatments (simple decoupled signal) highlighting its abundance in the species (Fig. 3).

Figure 3 NMR Spectra of NaH13CO3 incorporation into photosynthetic products of H. scabra in different light and darkness treatments (n= 3).

(A) 24 h at saturation irradiances; (B) 12 h under light saturation and 12 h in darkness; (C) 24 h in darkness. An asterisk (*) indicates consistent signals for aspartate in control treatment; two asterisks (**) indicate 13C enrichment (multiple coupling). A lowercase letter a indicates control, and a lowercase letter b indicates treatment.

Discussion

The results of our work support the hypothesis that a synergistic increase in pH, temperature, and nutrients enhances H. scabra photosynthesis. An increase in temperature could enhance Pgross at high pH if there are sufficient nutrients. Environmental conditions of high seawater temperature (Robledo & Freile-Pelegrín, 2005; Rodríguez-Martínez et al., 2010), alkaline pH seawater, likely because of the karstic origins of the Yucatán Peninsula (Cejudo et al., 2020), and pulsatile nutrient enrichment due to the submarine groundwater discharge (Hernández-Terrones et al., 2015), are common in Quintana Roo coastal areas where H. scabra and other Halimeda species colonize shallow environments. Interestingly, in the control treatment (no added nutrients) at low pH and high temperature high Pgross was also observed, most probably because at low pH CO2 availability is higher (Falkowski & Raven, 2007) and an increase in temperature facilitates photosynthesis (Bruno, Carr & O’Connor, 2015; Vásquez-Elizondo & Enríquez, 2016). Our results also emphasize that interactive effects are more reliable indicators than those observed under individual analyses since pH alone had no effect on Pgross while all interactions were significant. Variation of only one factor can modify the photosynthetic response to other factors highlighting the importance of interactive studies.

Conversely, the interactive effect of decreasing pH (low and medium) with increases in temperature and nutrient enrichment kept Pgross below its potential capacity. Thus, potential deleterious effects on H. scabra performance are expected to occur under future scenarios of ocean acidification, global warming, and their complex interactions with nutrient enrichment due to the continuous coastal development in the area.

In agreement with our results with H. scabra, significant reductions in gross photosynthetic rates have been reported for H. macroloba and H. cylindracea when exposed to elevated CO2 combined with elevated temperature, showing an additive negative effect (Sinutok et al., 2012). In contrast, for H. incrassata, H. simulans, and H. opuntia no significant effects in net photosynthesis were reported for the interactions among species, pH, and temperature (Campbell et al., 2016). While, in H. opuntia no interactive effect of CO2 and nutrient enrichment on net photosynthesis was found (Hofmann et al., 2015).

These contrasting results among congeners indicate that the photosynthetic responses to the interactive effects of several environmental variables are complex since, in addition to the factors being evaluated, the physiological mechanisms could be responding to other interrelated processes that were not assessed during assays. For example, Campbell et al. (2016) found in three Halimeda species that photosynthesis was positively correlated to calcification rates and, an increase in temperature increased activity of both processes. In this context, processes with high carbon requirements such as calcification could indirectly stimulate photosynthesis (Carvalho & Eyre, 2017), generating protons that are used to facilitate the absorption of nutrients and bicarbonate (McConnaughey & Whelan, 1997). The reduction of NO3− to NH4+ is another process with high energy requirements (Ale, Mikkelsen & Meyer, 2011), and it is related to carbon fixation (Cabello-Pasini & Figueroa, 2005) so it is likely to be more plausible to affect photosynthesis rather than calcification since the latter appears to be more dependent on photosynthetic activity of many calcifying primary producers. Nutrient enrichment supported a rapid increase in the physiological performance of H. opuntia (Teichberg, Fricke & Bischof, 2013). Therefore, the photosynthetic increase found with the nutrient addition (KNO3:K3PO4) in our experiments could be the result of its effect on processes related to nutrient uptake and these, to photosynthesis. Moreover, it is also known that nutrient uptake rates increase with temperature increases (Harrison & Hurd, 2001), consequently, our results not only are a response to the interactive effect of environmental factors but also, the result of the direct and indirect response of other metabolic processes on photosynthesis.

Temperature is a significant factor controlling metabolic rates, including photosynthesis; increasing temperature increases photosynthetic rates linearly up to an optimum rate, beyond this thermal threshold rates, tend to decline (Bruno, Carr & O’Connor, 2015; Vásquez-Elizondo & Enríquez, 2016). It is generally accepted that Q 10 values greater than 2 characterize an active nutrient absorption process across cell membranes, while Q 10 ∼1 values describe passive processes that are not greatly affected by temperature (Lobban & Harrison, 1994). According to this, our calculated Q 10 Pgross for the medium and high nutrient treatments are in the range of active nutrient absorption, expected by organisms living in highly illuminated habitats and with high elevated metabolic activity (Vásquez-Elizondo & Enríquez, 2016).

Mechanisms of photosynthetic carbon uptake can influence the isotopic composition of organic matter. Values of δ13C between -30 and -10‰ indicate active uptake of both HCO3− and CO2 and species whose fixation fall within this range are classified as species with active CCM (Maberly, Raven & Johnston, 1992; Raven et al., 2002; Diaz-Pulido et al., 2016; Bender-Champ, Diaz-Pulido & Dove, 2017). Species with δ13C signatures between −32‰ and −22‰ are considered as C3 plants while δ13C between −16‰ and –10‰ are typical for C4 plants (Valiela et al., 2018). Considering these ranges and the results obtained in this work, H. scabra could be classified as a C3 plant with a CCM that uses both, HCO3− and CO2 as a resource of Ci for photosynthesis. The δ13C values of H. scabra found within this study are in the range of those reported in other Halimeda species, such as H. opuntia, ∼−21‰ (Zweng, Koch & Bowes, 2018) and H. tuna, ∼−21‰ (Duarte et al., 2018).

The extracellular CA inhibitor (AZ), did not show any adverse effect on photosynthesis, indicating a direct uptake of HCO3− while a significant reduction in Pmax with EZ, confirmed HCO3−uptake and the presence of a CCM (Badger et al., 1998) along with the role of internal CA (Badger & Price, 1994). The reduction of photosynthesis under the activity of EZ was only about 22.2% relative to control samples, likely because there was still enough CO2 in the proximity of RuBisCO to maintain a reduced level of photosynthesis. The available CO2 may come from the following alternatives: (1) as the result of a CCM, mainly related to efficient HCO3− utilization (Raven, 1997); (2) respiratory CO2, (Borowitzka & Larkum, 1976); (3) CO2 supplied from interutricular spaces, although this source is not sufficient to sustain photosynthesis (De Beer & Larkum, 2001); and (4) by CO2 diffusion from both, the external medium and the intercellular space (ICS) (Borowitzka & Larkum, 1976). All previous explanations could maintain RuBisCO CO2 saturated and minimize photosynthetic losses after inhibiting intracellular CA. This suggests that inorganic carbon supply for photosynthesis in H. scabra does not depend entirely on the activity of the CA’s, and might be maintained by several mechanisms which may be advantageous during adverse conditions. Bicarbonate (HCO3−) uptake has also been found in H. discoidea, H. macroloba, and H. tuna (Borowitzka & Larkum, 1976), while the lack of extracellular CA has been found in H. discoidea (De Beer & Larkum, 2001) and H. cuneata f. digitate (Hofmann et al., 2015a).

Some Halimeda species possess CCM and use bicarbonate as an alternate source of inorganic carbon (Borowitzka & Larkum, 1976; Price et al., 2011) keeping their photosynthesis carbon saturated under the current seawater Ci concentrations (Beer, 1994; Beardall, Beer & Raven, 1998; Fernández, Hurd & Roleda, 2014). Therefore, this ability may also be responsible for Pgross enhancement under elevated temperature observed in this study in H. scabra through a decrease in photorespiration (Ogren, 1984). According to Giordano, Beardall & Raven (2005), the number of resources that a cell invests in acquiring carbon through a CCM is likely to be coupled with the availability of nutrients. This could also explain the increase in Pgross observed at the highest nutrient level since most CCM’s require the de novo synthesis of specific proteins, which represents a demand for cellular nitrogen (Giordano, Beardall & Raven, 2005). The low oxygen production observed at low pH (under high nutrient concentrations) in H. scabra could be a delay in the induction of the CCM relying on passive diffusion of CO2 alone, thus leading to reduced efficiency of carbon assimilation (Price et al., 2011; Cornwall et al., 2012; Meyer et al., 2016).

In this study, H. scabra incorporated 13C isotope into aspartate in the three photoperiod treatments demonstrating photosynthetic and non-photosynthetic NaH13CO 3assimilation. Moreover, the incorporation of 13C isotope in aspartate in 24-hour darkness treatment indicates β-carboxylation which facilitates a metabolic alternative to inorganic carbon carboxylation resulting in an important contribution to CCMs (Raven & Osmond, 1992; Enríquez & Rodríguez-Román, 2006). β-carboxylation has multiple functions for algal metabolism such as providing essential compounds for growth that cannot be produced photosynthetically (Falkowski & Raven, 2007). Carbon fixation of these compounds can be done in both light and darkness (Axelsson, 1988), and it is generally less than 5% of maximum photosynthesis (Cabello-Pasini & Alberte, 1997). In marine algae, the end products of this carbon fixation independent of light are typically organic compounds and amino acids rather than triose sugars generated during photosynthesis (Cabello-Pasini & Alberte, 1997). Although the δ13C values in H. scabra suggest a C3 pathway, the abundance of aspartate in control and experimental treatments also suggest the existence of a C4 pathway. In C4 plants, the C4 acids malate and aspartate are the major initial photosynthetic products, these products are rapidly decarboxylated releasing CO2 for its refixation by RuBisCO functioning as photosynthetic intermediates (Holaday & Bowes, 1980). In this sense, a C4 mechanism could explain the increase in Pgross under all of our high-end treatments (temperature, pH, and nutrient concentration), as well as the insensitivity of the photosynthetic response of the alga to AZ and, its low inhibition in the presence of EZ. C4 plants have an active CCM, which is mainly related to the efficient use of HCO3− through an initial carboxylation reaction by a Phosphoenolpyruvate enzyme (Badger & Price, 1994; Raven, 1997). C4 mechanisms have been reported in some Chlorophyta, including the semi-calcified Udotea flabellum, which shows an initial carboxylation by phosphoenolpyruvate carboxykinase enzyme (Reiskind, Seamon & Bowes, 1988), whereas in Ulva prolifera evidence of both C3 and C4 pathway has been found (Xu et al., 2012).

Conclusions

H. scabra uses both CO2 and HCO3− for photosynthesis and it seems to have different mechanisms for Ci acquisition incorporating bicarbonate through photosynthetic and non-photosynthetic pathways. Our results suggest the presence of both C3 and C4 pathways, with the latter relying on β-carboxylation. These strategies give H. scabra physiological plasticity to acclimate to possible environmental changes in the short term. Our study strongly suggests that H. scabra acclimatizes better to environmental conditions of high pH and high temperature with enough nutrient enrichment. Although these conditions could exacerbate the presence of epiphytes and opportunistic algae, the availability of pulsatile nutrients likely plays a role in maintaining Halimeda populations by enhancing algal photosynthetic performance. Such conditions are typical in the Yucatan peninsula coast where Halimeda species grow in abundance. Opposite interactive conditions of decreasing pH in combination with increases in temperature and nutrient availability, could keep photosynthesis at a sub-optimal level which has strong ecological implications due to the potential for decline in Halimeda abundance and the resulting consequences to sediment production and carbon balance in coral reefs where these algae thrive.

Supplemental Information

Supplemental Information 1 Experiment design

Experiment design to test the interactive effects of temperature, pH, and nutrients on gross photosynthesis of H. scabra.

Click here for additional data file.

Supplemental Information 2 Statistical analysis

Results of three-way ANOVA and Post hoc Newman-Keuls.

Click here for additional data file.

Supplemental Information 3 Photosynthesis data

Raw Halimeda scabra data

Click here for additional data file.

The authors thank Dr D Valdés, V Avila Velázquez, and C Chávez for their help in setting up the experiments and to Dr E Serviere-Zaragoza for the tissue isotopic analysis.

Additional Information and Declarations

Competing Interests

Author Contributions

Data Availability

The authors declare there are no competing interests.

Daily Zuñiga-Rios conceived and designed the experiments, performed the experiments, analyzed the data, prepared figures and/or tables, authored or reviewed drafts of the paper, and approved the final draft.

Román Manuel Vásquez-Elizondo analyzed the data, prepared figures and/or tables, authored or reviewed drafts of the paper, and approved the final draft.

Edgar Caamal performed the experiments, prepared figures and/or tables, and approved the final draft.

Daniel Robledo conceived and designed the experiments, analyzed the data, authored or reviewed drafts of the paper, and approved the final draft.

The following information was supplied regarding data availability:

Raw data are available in the Supplemental Files.

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
