# Peer review of "Photosynthetic responses of Halimeda scabra (Chlorophyta, Bryopsidales) to interactive effects of temperature, pH, and nutrients and its carbon pathways"

_PeerJ, doi:10.7717/peerj.10958_

## Round 0.1 · original submission · Minor Revisions

I now have three reviews from experts in photosynthetic physiology of marine plants, and their consensus is positive. All three provide comments for improving the manuscript, which the authors should address in a point-by-point manner in preparing their revision.
I look forward to seeing the revised manuscript.

·

Basic reporting

This ms reports on the results of a physiological investigation on the interactive effects of temperature, pH and nutrients on photosynthetic characteristics in the tropical calcareous green algae Halimeda scabra. The paper is general well written, logically-structured and well referenced. The introduction provides abundant background information on previous related studies and on the importance of Halimeda to tropical and subtropical reef systems. The introduction also does a good job of setting up the rationale for conducting this particular study and why they examined the 3 abiotic variables of temperature, pH and nutrients. The additional work investigating carbon acquisition mechanisms was also well presented and added to the discussion on possible mechanism responsible for the Pgross responses. The figures and tables are clear and well described.

Experimental design

The authors use a clear 3-way factorial design, which they specifically describe in Table S1. The only major question I have about their methods involves their stated measurement of respiration in the light. How was this done? Since other photosynthetic and respiration measurements were accomplished by measuring oxygen fluxes, the authors need to specify how they could measure respiration in the light, which is not possible with oxygen flux. The authors also need to refer to their nutrient treatments based on relative concentrations (low, medium and high), not on ratios, which are constant at 10:1.

Validity of the findings

The statistical analyses of the experimental data were straightforward, clearly presented and appropriate to the authors’ objectives. Table S2 clearly shows which main effects and treatment combinations were significantly different from one another. The authors do need to include more text in the results and discussion regarding the lack of a significant pH effect as this is a novel finding. Also, the control (no added nutrients), @low pH and high temperatures had the 2nd highest Pgross, which the authors do not discuss, although they do point this out in the results. This warrants some discussion. In Fig. 1 it appears that at low pH, an increase in nutrients reduces Pgross across temperatures while at high pH, nutrients increase Pgross. These patterns need to be more fully discussed and possible explanations for the patterns need to be included.
The treatment of the carbon acquisition stable isotope and CA inhibitor data in the results and discussion is clear and well presented.

Additional comments

This paper presents valuable new data on the ecophysiology of an important macro-algae species. With sufficient revision, addressing the comments above and those included in the annotated pdf, this paper should be suitable for publication. Primarily, the similarity in Pgross responses between the control, low pH treatment plants and the high nutrient, high pH treatment plants with increasing temperatures needs to be acknowledged and discussed.

·

Basic reporting

The manuscript is well written and easy to follow. The ms provide general information about the topic covered. It is believed to be current and accurate, I would suggest to add some important references in the topic of carbon concentrating mechanisms in algae.

*Raven, J. A. 1997. Inorganic carbon acquisition by marine autotrophs. Adv. Bot. Res. 27:85–209.
*Beardall, J., Beer, S. & Raven, J. A. 1998. Biodiversity of marine plants in an era of climate change: some predictions based on physiological performance. Bot. Mar. 41:113–23.
*Fernandez, PA 2014. Bicarbonate uptake via an anion exchange protein is the main mechanism of inorganic carbon acquisition by the giant kelp Macrocystis pyrifera (Laminariales, Phaeophyceae) under variable pH. J Phycol. 50(6):998-1008.
*Beer, S. 1994. Mechanisms of inorganic carbon acquisition in marine macroalgae (with special reference to the Chlorophyta). Progr. Phycol. Res. 10:179–207.

Experimental design

Research questions are very well defined. However, I have some questions about the experiment design:

On line 121: The authors mentioned that n was = 8 bottles (I assumed this is one bottle per light treatment plus a blank). What about replication? how many replicates per treatment combination? You have 3 pHs x 3 Temp x 4 Nutrients ratio x ? replicates. They did not mention replication number per treatment combination. With a minimum of three replicate per treatment I got a total of bottles of 108. So, I would like to know how the authors carried out the photosynthetic measurements (e.g., all samples were carried out at the same time/day/hours?).

On line 124: The authors mentioned that the temperature was maintained constant, but they do not mention at what temperature. Were all experimental treatment combinations (pH, nutrients ratio and temperature) carried out at the same time? they do not mention how the experiments were perform. This information is very relevant to interpret the results.

On line 139: Which is the reference for the seawater pH manipulation?
On line 170: How the seawater was sterilised (filter, autoclaved)?
O line 175: Why the authors used a different control temperature in this experiments (24°C vs 28°C)?
On line 182: Light saturation of 278 umol, How was this intensity estimated?

Results:

Figure 1: why the authors used a line graph instead of e.g., bar graphs ? were the experiments continuous over time? Did they used the same piece of tissue across the experimental temperatures? Also, why the graphs are separated by pH? were the experimental treatments carried out at different times? I would suggest to modify this figure and include the n replication in the legend.

Discussion:

On line 303: a CCM such us?

Validity of the findings

The results are novel and interesting for the specie, and conclusions are well stated according to the results.However, my main concern is about the replication and how the experiments where carried out. The authors will need to clarify this in the ms, previous acceptance of the ms.

Reviewer 3 ·

Basic reporting

no comment

Experimental design

no comment

Validity of the findings

no comment

Additional comments

The manuscript "Photosynthetic responses of Halimeda scabra (Chlorophyta,
Bryopsidales) to interactive effects of temperature, pH, and nutrients and its carbon pathways",in the study, it was divided into two parts, first part, it is about the effect of environmental factors on photosynthstic response of Halimeda scabra, second part, it is about the carbon pathways. Here are some major questions in the study, 1)please clarify the relationship between these two parts, the temperature is about 24℃ in field, and highest temperature in this sampling site is around 30℃, Why did you set highest temperature as 33℃, did you consider the temperature condition in future? Why all assays conducted at 28℃ in second part, did the condition (temperature and light saturation) from the first part experiment results? 2) which treatment is the basic control treatment in the experimental design,main effects and and interactive two of them on photosynthetsis, similar positive effect also caused by the interactive effect between high temperature and low pH. Why all didn’t be discussed?
I believe that study includes some meaningful results on the effects of photosynthesis for Halimeda scabra and its carbon pathways. So, I suggest that the authors revise the manuscript carefully and resubmit to PeerJ.
Here are some specific comments. Hopefully, it will help you to improve ms.

Specific comments:
English: proofread is needed.
Abstract:
P5 L19, 25: define Pgross and CCM for the first time in abstract
Introduction:
P7 L90:which kind of temperature (high or low) had positive synergistic effect in reference?
P7 L102: “nutrient ratios” change into “nutrient levels”
P8 L136-145:the content of experimental design move to line 115
P8 L136: How about the concentration of HCl and NaOH?
P8 L136: the ratio of nutrient is same in the experimental design, all “Nutrient ratio” suggest to change into “nutrient levels”
P9 L175: Why did you conducted carbonic anhydrase inhibition assays at 28 ℃,not the current cultivation temperature of 24℃?
P9 L182: In this experiment, there is no contents showed that the light saturation is 278 μmol photons m-2 s-1, how can you get it.
P10 L211: “p=0.0000” change into “p<0.001”
P10 L212-214: the result is too simple, is there major effect or interactive effects between two of them on photosynthetic? Show no details, and clarify the topic of sentence.
P10 L218-223: here described the effect of nutrients and pH effects on Pgross, the title is only show the temperature effects, it would be better clarify the background of nutrient and pH ?
Figure legends:
P 20 L582:Please add the number of duplicate samples,“n=?”
P23 Figure 3:Complete the figure notes
P26 Tables:“Effect of temperature on P gross (Q 10 ) in three nutrient concentrations (Two Way-ANOVA)” change into “Effect of temperature on P gross (Q 10 ) in three nutrient concentrations and three pH levels (Two Way-ANOVA).”

---

## Round 0.2 · accepted · Accept

The authors have satisfactorily addressed the comments of the reviewers in preparing their revised manuscript. I can now recommend its publication to the editorial staff.